# Case Series of Precision Delivery of Methylprednisolone in Pediatric Inflammatory Bowel Disease: Feasibility, Clinical Outcomes, and Identification of a Vasculitic Transcriptional Program

**DOI:** 10.3390/jcm12062386

**Published:** 2023-03-20

**Authors:** Steven Levitte, Reza Yarani, Abantika Ganguly, Lynne Martin, John Gubatan, Helen R. Nadel, Benjamin Franc, Roberto Gugig, Ali Syed, Alka Goyal, K. T. Park, Avnesh S. Thakor

**Affiliations:** 1Interventional Radiology Innovation at Stanford (IRIS), 3155 Porter Drive, Palo Alto, CA 94304, USA; 2Division of Pediatric Gastroenterology, Hepatology, and Nutrition, Stanford University, Palo Alto, CA 94304, USA; 3Department of Pediatric Radiology, Interventional Radiology, Stanford University, Palo Alto, CA 94304, USA; 4Division of Gastroenterology and Hepatology, Stanford University, Palo Alto, CA 94304, USA

**Keywords:** inflammatory bowel disease, vasculitis, Crohn’s disease, locoregional, precision delivery

## Abstract

Systemic steroid exposure, while useful for the treatment of acute flares in inflammatory bowel disease (IBD), is associated with an array of side effects that are particularly significant in children. Technical advancements have enabled locoregional intraarterial steroid delivery directly into specific segments of the gastrointestinal tract, thereby maximizing tissue concentration while limiting systemic exposure. We investigated the feasibility of intraarterial steroid administration into the bowel in a cohort of nine pediatric patients who had IBD. This treatment approach provided symptom relief in all patients, with sustained relief (>2 weeks) in seven out of nine; no serious adverse effects occurred in any patient. In addition, we identified patterns of vascular morphologic changes indicative of a vasculopathy within the mesenteric circulation of inflamed segments of the bowel in pediatric patients with Crohn’s disease, which correlated with disease activity. An analysis of publicly available transcriptomic studies identified vasculitis-associated molecular pathways activated in the endothelial cells of patients with active Crohn’s disease, suggesting a possible shared transcriptional program between vasculitis and IBD. Intraarterial corticosteroid treatment is safe and has the potential to be widely accepted as a locoregional approach for therapy delivery directly into the bowel; however, this approach still warrants further consideration as a short-term “bridge” between therapy transitions for symptomatic IBD patients with refractory disease, as part of a broader steroid-minimizing treatment strategy.

## 1. Introduction

Inflammatory bowel disease (IBD), comprised predominantly of Crohn’s disease (CD) and ulcerative colitis (UC), is characterized by an inappropriate immune response creating inflammation at intestinal mucosal, transmural, and extra-intestinal locations. Treatments focus on achieving mucosal healing through the use of anti-inflammatory therapies delivered orally or intravenously. Corticosteroids are commonly used in active IBD to achieve clinical remission as a bridge to long-term steroid-free therapy. However, steroid therapy exposes patients to a variety of side effects, which is particularly true in pediatric patients who are more likely to also experience refractory and steroid-dependent disease [1]. Given these clinical challenges, we hypothesized that, in the appropriate clinical indications, locoregional steroid administration into the mesenteric vasculature (i.e., precision delivery to the gastrointestinal tract) could be a viable strategy for achieving short-term clinical remission in children while minimizing systemic side effects. The feasibility of intraarterial therapy delivery directly into the bowel of patients with active IBD remains an open question, particularly in light of mesenteric angiogenesis, which has been described in IBD [2,3].

In children with first-line refractory IBD (either primary or secondary biologic non-response), we used intraarterial methylprednisolone to temporarily abrogate localized inflammation in order to establish second-line biologic therapy or to alleviate obstructive symptoms from intestinal strictures. In this pilot study, we report temporary improvement in nine out of nine patients treated, with responses lasting up to six weeks in some patients, with no major complications. We additionally describe findings of a mesenteric vasculopathy which we identified in pediatric patients with clinically active CD using high-resolution imaging and which appears to be independent of the already documented process of mesenteric angiogenesis [4]. These findings were used to inform the analysis of endothelial transcriptional profiles in patients with active CD leading to the expression change in specific vasculitis-associated genes that have not been previously implicated as potential IBD risk foci.

## 2. Materials and Methods

### 2.1. Patient Selection and Chart Review

We included patients with active CD (defined as Pediatric Crohn’s Disease Activity Index ≥ 20) or UC (defined as Pediatric Ulcerative Colitis Activity Index ≥ 30) at an academic referral center between 1 October 2021 and 1 January 2023, according to an institutional multi-consensus “escalation-of-care” protocol with IRB approval (IRB 64154). Patients with an established diagnosis of IBD who had symptoms of clinical flare were referred by their primary attending gastroenterologist to interventional radiology. The exclusion criteria were age < 4 years, MRI-unsafe metallic implants, contrast allergy, and impaired renal function with GFR < 45 mL/min. Formal angiography with targeted methylprednisolone administration is performed preferably at the same time as MRI, but at most within one week. Chart review was conducted for 60 days following the procedure to determine the clinical response and adverse events including pain at the arterial access site, bleeding, hemangioma, aneurysm, and site infection.

### 2.2. Imaging

Magnetic resonance enterography (MRE) was performed as part of a standard-of-care approach and included early arterial phase imaging that is equivalent to a contrast-enhanced MRA (CE-MRA) at our institution. Patients were instructed to drink 20 mL/kg of a mannitol-based oral contrast agent (Breeza, Beekley Medical, Bristol, CT, USA) prior to imaging. MR enterography was then performed, consisting of coronal and axial T2-weighted SSFSE, axial diffusion weighted imaging, coronal multiphase T2 SSFSE for peristalsis assessment, coronal pre-contrast and dynamic post-contrast T1 weighted fat-suppressed FSPGR, and axial post-contrast T1 weighted imaging with two-point Dixon fat suppression. Immediately prior to post-contrast imaging, patients were administered 0.5–1 mg of intravenous glucagon to reduce the motion artifact from peristalsis. Dynamic post-contrast imaging included at least two arterial phases [5] and an enteric phase after intravenous administration of standard dose gadobenate dimeglumine (MultiHance, Bracco Diagnostics, Monroe Twp, NJ, USA). For patients in this study, prior to glucagon or full contrast dose administration, a test bolus of contrast was administered to determine the optimal arterial timing for post-contrast imaging [6]. Thus, the first phase of the post-contrast imaging acquisition resulted in a contrast-enhanced MRA, which was used to assess vessel course and morphology. In two patients, the quality of initial gadolinium-based CE-MRA was determined to be inadequate. As part of a multidisciplinary consensus for “escalation-of-care” at our institution, these patients were administered intravenous ferumoxytol (Feraheme, AMAG Pharmaceuticals, Waltham, MA, USA), and repeat coronal T1-weighted imaging was performed [7,8]. Exams were interpreted by an attending radiologist with experience in abdominal imaging (ABS), in accordance with consensus guidelines [9]. In addition to standard reporting, the vascular territory of each diseased segment was included in the interpretation. In patients with non-diagnostic conventional CE-MRA, supplemental CE-MRA was performed immediately prior to digital subtraction angiography (DSA). The DSA images were compared with the MRI angiogram in real time to confirm vessels supplying diseased segment(s) of the bowel.

### 2.3. Methylprednisolone Therapy Administration

Patients were placed supine on the angiography table, and both groins were prepped and draped in the usual sterile fashion. Under ultrasound guidance, the left/right common femoral artery was accessed, and a 4Fr sheath was inserted. Following the insertion of the sheath, heparin was given at 50 IU/Kg. The superior mesenteric artery (SMA) or inferior mesenteric artery (IMA) was cannulated, and an angiogram was performed using contrast medium injection. The angiogram was compared with the MRI angiogram, and the vessels that supplied the diseased segment of the bowel were identified. Using a microcatheter system, these vessels were then selectively cannulated. Methylprednisolone was delivered (60–100 mg, body weight-dependent, Table 1) over 20 min. This procedure was then repeated for other diseased segments of the bowel, as indicated. At the end of the case, the catheters and sheaths were removed from the patient, and hemostasis was achieved using manual compression for 10–15 min.

### 2.4. Transcriptomic and Single Cell Analysis

For transcriptomic and single-cell experiments, we identified microarray-based studies in the NCBI Gene Expression Omnibus (GEO) database, which included active and inactive CD cases. The included studies were GSE6731, GSE16879, GSE59071, and GSE67106. We included vasculitis and angiogenesis-associated genes if they were detected by at least two of the four studies. We downloaded a single-cell dataset (GSE134809) which included ten patients with ileal CD who underwent ileal resections and had paired tissue obtained from inflamed and non-inflamed ileum. The datasets were merged, and normalization, scaling, dimensionality reduction, and clustering were performed. We identified a cluster of CD31 (PECAM–1)-positive cells which were designated as endothelial cells.

### 2.5. Statistics and Data Presentation

Graphs were made using Prism9. Linear regression was performed using Prism9 (Graphpad Software). The significance of differential expression was assessed by an adjusted *p* value < 0.05 in at least one study. Statistical significance was assessed between groups for clinical significance using a paired t-test in Prism9. The mean expressions of vasculitis- and angiogenesis-associated genes were compared between endothelial cells and inflamed versus non-inflamed ileum using a Wilcoxon rank-sum test. Statistical significance was defined as * *p* < 0.05, ** *p* < 0.01, *** *p* < 0.001, **** *p* < 0.0001.

## 3. Results

### 3.1. Clinical Outcomes

A total of nine patients with IBD were treated with locoregional methylprednisolone (Table 1). In this cohort, six patients had CD and three had UC. All patients had previously been treated with systemic corticosteroids due to disease severity, and all had experienced treatment failure with at least one biologic agent. Seven patients experienced moderate to marked clinical improvement, while two others had mild transient symptom improvement. Those with substantial clinical responses included patients with Crohn’s intestinal fibrosis (3), refractory Crohn’s colitis (2), and UC refractory pancolitis (2). The patients who transiently responded had diagnoses of Crohn’s intestinal fibrosis (1) and UC refractory pancolitis (1), and in these patients, intraarterial steroids were used more for a salvage therapy during their flare episode. Of the Crohn’s intestinal fibrosis cases (4), three underwent endoscopic dilation, which was successful in two cases; the third underwent resection for a refractory duodenal stricture. A paired t-test performed using PCDAI scores collected before and after intraarterial methylprednisolone in patients with CD demonstrated statistically significant improvement (*p* = 0.025), although the response duration was variable. Of the three patients with UC, two experienced marked improvement, although the difference in PUCAI scores was not significant in aggregate (*p* = 0.19). Of the five patients treated as “bridge” therapy (with the intent to induce short-term clinical remission while undergoing second-line biologic dose optimization), four experienced moderate to marked sustained clinical improvement.

### 3.2. Feasibility and Adverse Events

All of the patients in this cohort successfully underwent MRE followed by mesenteric angiography and intraarterial steroid administration. There were no non-diagnostic MRE examinations. MRA and DSA morphology assessment was completed successfully in all patients. There were no adverse events from arterial access, including bleeding at the arterial access site, hemangioma, aneurysm, or site infection. One patient reported discomfort at the access site during a clinic visit one week following the procedure.

### 3.3. Vascular Findings in Patients with CD

During the course of patient imaging, which included MRI and intraarterial steroid administration, we observed abnormal vascular morphology, which we then further investigated by re-analyzing imaging obtained at the time of treatment for five patients with CD who had Pediatric Crohn’s Disease Activity Index (PCDAI) scores ranging from 20 to 37.5. Magnetic resonance enterography (MRE) demonstrated active bowel inflammation characterized by segmental mural enhancement and mural edema in all five patients (Figure 1A–E and Table 2) [10]. The two patients with the highest PCDAI scores (Figure 1A,B) underwent colonoscopy immediately prior to imaging and angiography, which was notable for extensive mucosal friability and exudate consistent with severe Crohn’s colitis. Mucosal biopsies reviewed as part of standard-of-care by a clinical pathologist noted active mucosal inflammation, but the microvascular structure was normal. Vascular mapping using contrast-enhanced magnetic resonance angiography (CE-MRA) showed markedly engorged vasa recta supplying areas of an actively inflamed bowel, consistent with findings in adults with active Crohn’s disease [11]. In three patients who had moderate to severe active disease (PCDAI 35–37.5), marked tortuosity of the vasa recta was appreciated (Figure 1A–C). A similar but less prominent phenotype was observed in one patient with moderate but persistent disease (Figure 1D). In the final patient who had mild but treatment-resistant disease, the vasa recta were engorged but had a predominantly linear and slightly pruned appearance (Figure 1E). In all cases, abnormal vasculature was observed only in the affected bowel segments with evidence of active inflammation, and none of the patients had clinical evidence of systemic vasculitis. Subsequent digital subtraction angiography (DSA) confirmed abnormalities of the vasa recta in all patients (Figure 1F–J). In the three patients with the highest clinical scores, the DSA of the involved areas of the bowel demonstrated hyperemia and dysplastic-appearing small- and medium-sized vessels with tortuosity and beading/irregularity of the arteries with demarcated strictures (Figure 1F–H). Mural engorgement and contrast retention with reduced clearance in the delayed phase of angiography were observed. In contrast, the two patients with lower disease activity (Figure 1I–J) demonstrated an increased number of pruned and linear vasa recta, but without the tortuous, beaded morphology of the more active patients. We developed a scoring system to assess the degree of vascular tortuosity, beading, stricture, and occlusion. MRA and DSA images from each patient were assessed blinded to the PCDAI score. MRA and DSA scores both correlated with PCDAI, although DSA displayed a higher coefficient of determination (DSA r^2^ = 0.79, MRA r^2^ = 0.24, Figure 1K). The patients with higher scores by DSA tended to have a more pronounced clinical response, although the number of cases was insufficient for further statistical analysis.

### 3.4. Transcriptomic Analysis of Vasculitis-Associated Genes in CD

Vascular angiogenesis and endothelial dysfunction have both been well-described in patients with CD, but our detailed morphological analysis suggested the presence of vasculitis as a feature of actively inflamed regions of the bowel. We therefore asked whether unique alterations in vasculitis-associated molecular pathways (as opposed to angiogenesis-associated pathways) may be present in those patients with active CD. We used the QIAGEN Ingenuity Pathway Analysis program to identify genes associated with vasculitis, angiogenesis, or both. From the Gene Expression Omnibus (GEO) database, we identified studies including active and inactive CD and non-IBD controls (131 patients total, 49 with active CD, 47 with inactive CD, and 35 non-IBD patients). Of 102 vasculitis-associated genes, 60 (59%) were significantly (*p*-value < 0.05 adjusted for multiple comparisons) differentially regulated in active CD, while 22 (22%) were significantly differentially regulated in inactive CD. Of 1181 angiogenesis-associated genes, 679 (57%) were differentially regulated in active CD, while 248 (21%) were differentially regulated in inactive CD. Of 154 genes implicated in both vasculitis and angiogenesis, 94 (61%) were differentially regulated in active CD, while 37 (24%) were differentially regulated in inactive CD. We created interaction maps to understand the molecular pathways implicated in the differentially expressed genes in active vs. inactive disease (Figure 2A–C).

### 3.5. Single-Cell Analysis of Vasculitis-Associated Genes in the Endothelium of Active CD

Next, we asked whether endothelial disruption could be correlated with expression changes in vasculitis-associated genes that we observed in patients with active CD. We analyzed the single-cell expression of the 60 differentially expressed vasculitis-associated genes in the endothelial cells of 10 patients with active CD who required bowel resection due to uncontrolled disease activity [12] (Figure 2D). Twenty-three genes showed strong expression in the endothelial cell subset; of these, the expression of nine was significantly different between inflamed and noninflamed areas of the bowel (Figure 2E). None of these genes had been implicated as a CD risk locus in genome-wide association studies [13]. Their functions include protecting endothelial integrity (CAV1, ABL2, FLT1), mediating vascular permeability (COL18A1), limiting injury-induced inflammation (TIMP3), and modulating coagulation (SERPINE1, PLAT) [14,15,16]. Together, these results implicate a subset of vasculitis-associated genes as being differentially expressed in the intestinal endothelium in active CD.

## 4. Discussion

While effective in achieving clinical remission, systemic steroid administration is associated with a wide array of side effects that are of particular concern for children [1]. We conducted a pilot study to evaluate the feasibility and potential efficacy of the locoregional intraarterial administration of methylprednisolone into the mesenteric vasculature of patients with severe, refractory IBD experiencing a clinical flare. From the imaging studies, we additionally characterized vascular patterning of the diseased segments of bowel in comparison to the non-inflamed areas in the same patient. All patients treated with intraarterial methylprednisolone experienced symptom improvement, with 2 or more weeks of effect in 78% of patients (7/9). There were no significant adverse effects associated with treatment in any patient. We leveraged MRE, CE-MRA, and DSA imaging, which have unprecedented resolution given recent technological advances, to describe morphologic changes in the mesenteric vasculature of pediatric patients with active CD. We analyzed large, publicly available transcriptomic studies, which confirmed a vasculitis-associated molecular signature unique to CD patients with active disease, and used single-cell expression analysis to identify genes involved with altered intestinal endothelial expression. Together, these findings suggest that the further study of vascular inflammation (i.e., vasculitis) as a driver of IBD pathogenesis could identify novel disease biomarkers and therapeutic targets.

The intraarterial steroid treatment of IBD was first reported in 1974, with the largest published cohort including 37 adult ulcerative colitis patients who received prednisolone via the superior and inferior mesenteric arteries; 57% demonstrated a marked or moderate treatment effect [17]. There have been no published reports of this treatment in a pediatric population, but it is interesting that the overall efficacy in children is the same, or better, given the frequently refractory nature of pediatric IBD [1]. Significantly, no major adverse effects were reported. Angiography is a well-established procedure, with over a million cases performed annually, with an overall complication rate (including the most common complications such as hematoma, vessel injury, and vessel thrombosis) below 0.5%; pediatric-specific data are more limited but are suggestive of similar rates [18,19,20,21,22]. To place this in context, pediatric endoscopy in high-risk patients (such as those with inflammatory bowel disease) is generally reported as 2–3% [23].

Our cases fell into two categories: children with primary or secondary biologic failure and acute inflammatory symptoms (rectal bleeding, diarrhea) who required short-term clinical remission while establishing therapy with second-line or third-line biologic agents; and patients with Crohn’s disease-associated intestinal stricture (who had symptoms of vomiting, weight loss, and/or abdominal pain). While two out of four patients with stricture improved and tolerated medical management (including successful endoscopic dilation following intraarterial steroid administration), the others either failed to achieve adequate clinical response or relapsed shortly after treatment and ultimately required surgical resection. The angiography and MRE findings were not substantially different between those who responded and those who did not; further work will be required to differentiate patients with fibrotic vs. inflammatory strictures, with the latter being more amenable to medical therapy. The clinical responses in patients with acute refractory colitis were more pronounced, with four out of five patients exhibiting moderate to marked clinical improvement, with a durability of response of roughly 2–3 weeks. Future work will aim to increase the number of patients in order to assess efficacy in patients with acute refractory colitis.

The pharmacokinetics of intraarterial vs. intravenous methylprednisolone administration remain unknown, although theoretical work suggests local drug concentrations may be five- to ten-fold higher compared to the intravenous administration of an equivalent dose [24]. Furthermore, glucocorticoids have both non-genomic (immediate) and genomic (delayed) immunomodulatory effects, which are relevant when considering pulse therapy vs. long-term systemic treatment [25]. In the setting of an IBD flare, patients are often treated with high-dose systemic steroids for 2–4 weeks, followed by a period of additional systemic treatment, which can last for eight weeks or more; this course is often even longer in patients who have biologic primary or secondary non-response due to the length of time required for second-line agents to have therapeutic effect [26]. We hypothesize that intraarterial steroid delivery could comprise a component of a steroid-minimizing treatment strategy for patients with acute refractory colitis by optimizing tissue drug concentration during pulse therapy and providing a longer-lasting anti-inflammatory effect, therefore enabling more effective dose de-escalation. We administered between 5 mg/kg and 10 mg/kg methylprednisolone depending on the disease severity and extent, with the highest dose being 450 mg, although many patients received substantially lower doses (Table 1) comparable to the dosing used in patients admitted for acute colitis. In a separate study of intraarterial therapy delivered to the liver in acute hepatic graft-versus-host disease, we have administered doses up to 1 g without notable adverse events. Further studies will be needed to determine the optimal dosing and speed of delivery [27]. Given the pilot data from this study, we propose further investigating the use of intraarterial methylprednisolone as a bridge therapy for primary or secondary biologic non-response in pediatric patients with the goal of reducing time to steroid-free remission; to this end, we have initiated a registered clinical trial (NCT NCT05587673).

This study also suggests a shared transcriptional program linking vascular inflammation and IBD, which has implications for future therapeutic avenues. In-depth imaging in our CD cohort revealed that the vascular changes are not limited to angiogenesis, which has been conventionally well-described in IBD [4]. The mesenteric vasculopathy seen in our CD patients is morphologically consistent with vasculitis, which aligns with molecular evidence pointing to broader vascular dysfunction in IBD from our work and others’ [28]. While anti-TNF therapy is increasingly used in the treatment of systemic vasculidities, selective vascular targeting could be an attractive new avenue for IBD therapeutic development. For instance, the inhibition of Abl2 has been shown to improve endothelial function in inflammatory states, but it has not been explored in the context of IBD [14]. Ongoing vascular inflammation could partially account for the poor correlation between serum and tissue anti-TNF levels seen in patients with active disease and could support endovascular locoregional delivery strategies, particularly in children, given that this approach would ensure endothelial cell exposure to therapies [17,29]. The locoregional arterial delivery of therapeutic agents that directly target endothelial dysfunction could therefore present an attractive therapeutic strategy which warrants further consideration. Imaging and molecular studies of tissue samples taken before and after these therapies can then help to determine how changes at the level of the endothelium can correlate with the overall clinical outcomes of patients.

Our study has some important limitations. First, this was a single-center pilot study with a small sample size that may be subject to sampling bias, despite our use of consecutive patients. Second, detailed molecular data were not available for the patients in our study, and whether the transcriptional changes identified in adult patients are present in pediatric patients remains unknown. Lastly, while the analysis of endoscopic biopsies from the two patients who underwent colonoscopy did not reveal alterations in arterial morphology, this analysis is limited by the depth of the tissue biopsy, which includes only the mucosal layer which contains very small vessels. Future studies will investigate whether morphologic changes are present in the larger arteries of the bowel wall in patients undergoing bowel resection.

In conclusion, we provide the first report of intraarterial methylprednisolone administration for the treatment of pediatric IBD, which was safe and well-tolerated in a small cohort of patients. This approach allows for the precision delivery of therapy into the bowel to facilitate local tissue exposure to therapeutic concentrations of therapy, while minimizing systemic and off-target tissue exposure. We describe morphologic evidence of mesenteric vasculopathy using high-resolution imaging techniques in pediatric patients with active CD, which correlates with evidence of the altered expression of vasculitis-associated genes within the intestinal endothelium of patients with active CD. Larger prospective studies are needed to validate these findings, assess efficacy in CD and UC, clarify the dynamics of vascular changes in the setting of inflammation in IBD (i.e., whether the changes result from intrinsic processes within the vessel itself or an extrinsic effect induced by the local microenvironment), and elucidate the mechanisms linking vasculopathy with IBD.

## Figures and Tables

**Figure 1 jcm-12-02386-f001:**
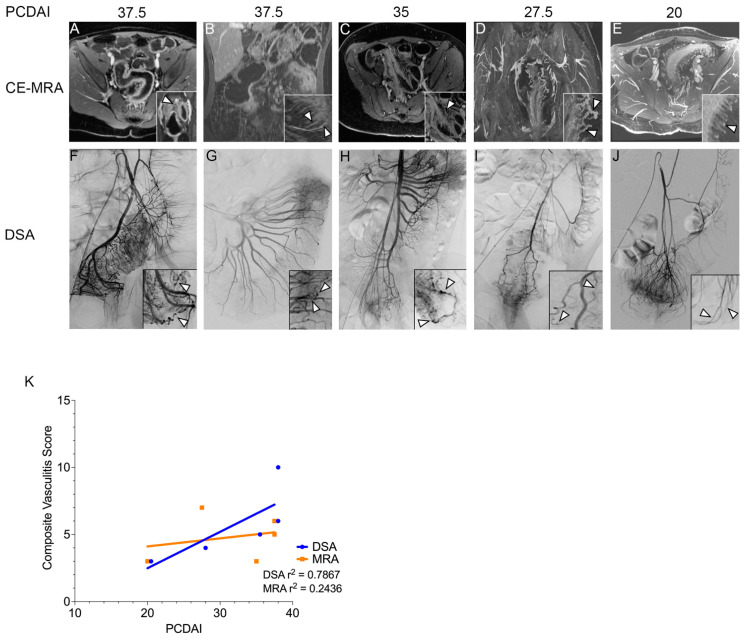
Vascular imaging of pediatric patients with Crohn’s disease. Contrast-enhanced magnetic resonance angiography (CE-MRA) (**A**–**E**) and digital subtraction angiography (DSA) (**F**–**J**). (**A**–**E**) show representative axial or coronal CE-MRA images. Arrowheads denote areas of altered vascular morphology including tortuosity (Panel (**C**,**F**) inset arrowheads), stenoses (Panel (**D**,**G**) inset arrowheads), and beading (Panel (**H**) inset arrowheads). Panel (**J**) inset arrows show angiogenesis seen in a relatively quiescent region of the bowel but without altered vascular morphology. Panel (**K**) shows linear regressions for vasculitis scores; MRA and DSA images were independently scored by two diagnostic radiologists who were blinded to the patient’s PCDAI score. Differences were resolved by consensus. Each image was assessed on a 0–5 scale of vascular tortuosity, beading, occlusion, and stricture; the composite score was generated as a sum of these four categories.

**Figure 2 jcm-12-02386-f002:**
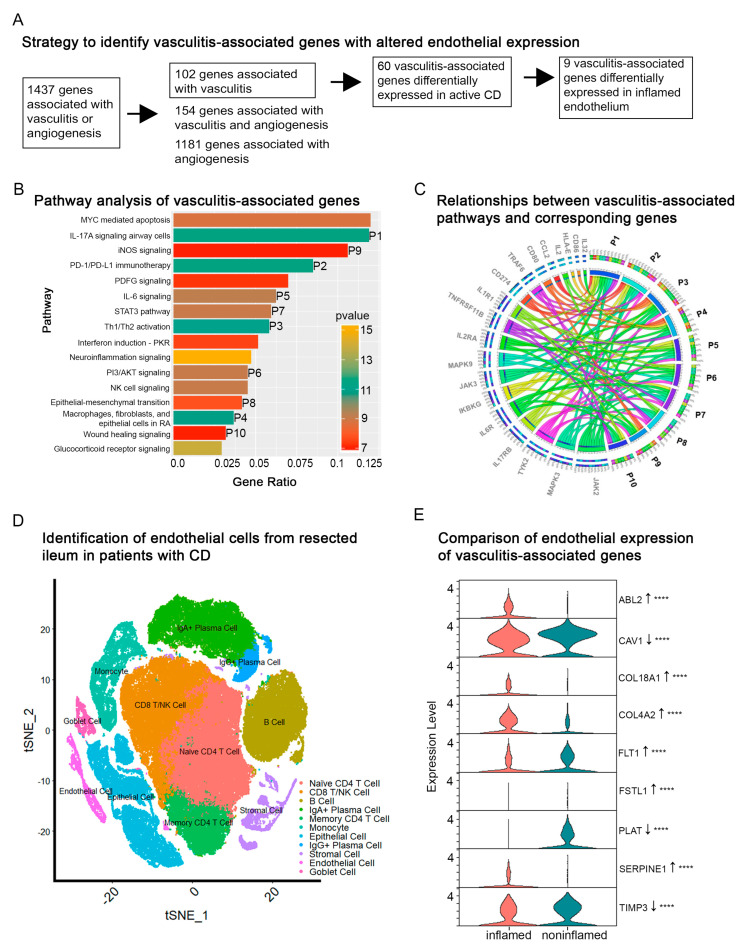
Transcriptomic and single-cell analysis of vasculitis and angiogenesis pathways in Crohn’s disease. Panel (**A**) shows the significant (*p*-value < 0.05) pathways identified among the 60 vasculitis-associated genes that were differentially expressed in patients with active CD. Panel (**B**) depicts the core pathway analysis which was performed using the Ingenuity Pathway Analysis (IPA) database. X-axis denotes the Gene Ratio (number of genes identified in our screen compared to the total number of background genes associated with the pathway) for each pathway; color-coding depicts the significance of each pathway based on its *p*-value. Panel (**C**) depicts a Circos plot showing the relationship between enriched vasculitis-associated pathways and their corresponding genes identified in our screen. Clockwise from top: pathways and genes are ordered by their number of interactions. Ribbon size encodes cell value associated with row/column segment pair. Column segment value and ribbon color are decided by the number of interactions but are not indicative of statistical significance. Panel (**D**) shows a t-distributed stochastic neighbor embedding (t-SNE) plot depicting immune and non-immune subsets of cells in 10 patients who had intestinal resection for active CD. We identified a cluster of CD31 (PECAM−1)-positive cells which were designated as endothelial cells. Panel (**E**) shows the relative endothelial expression level of nine genes that were significantly differentially expressed between inflamed and noninflamed areas of patients who underwent bowel resection for active CD. Arrows indicate the direction of altered expression. **** indicates *p* < 0.001.

**Table 1 jcm-12-02386-t001:** Clinical characteristics and outcomes of patients with IBD treated with locoregional methylprednisolone.

Patient Background	Age (y)	Sex	Weight (kg)	Pre-Procedure Disease Score	Vessel(s) Accessed and (Methylprednisolone Dose)	Post-Procedure Disease Score (Time)	Response Duration	Treatment Effect(s)
CD *								
Distal colitis	17	M	44	PCDAI 27.5	superior rectal artery (80 mg)	PCDAI 17.5 (1 week)	1 month	Resolution of abdominal pain, improvement in hematochezia
Pancolitis	10	M	24	PCDAI 37.5	superior rectal artery (50 mg), sigmoidal artery (50 mg)	PCDAI 7.5 (4 weeks)	Remains in remission	Resolution of hematochezia, resolution of abdominal pain; transitioned to adalimumab from vedolizumab
Intestinal fibrosis; ileal stricture	19	M	57	PCDAI 20	SMA: ileocolic branch (30 mg), ileal branch (50 mg)	PCDAI 0 (3 weeks)	Remains in remission	Resolution of abdominal pain, resolution of nausea; successful endoscopic dilation
Intestinal fibrosis; ileal stricture	17	F	73	PCDAI 37.5	ileocolic artery (50 mg), right colic artery (50 mg)	PCDAI 15 (2 days)	2 weeks	Resolution of abdominal pain, resolution of vomiting; redeveloped obstruction
Intestinal fibrosis; jejunal stricture	17	F	58	PCDAI 35	SMA: branch vessels supplying jejunum (300 mg)	Undetermined (4 weeks)	Remains in remission	Resolution of nausea and vomiting, abdominal pain unchanged; successful endoscopic dilation; transitioned to ustekinumab from adalimumab and methotrexate
Intestinal fibrosis; duodenal and jejunal strictures	17	M	45	PCDAI 32.5	gastroduodenal artery and branches supplying jejunum (270 mg)	PCDAI 30 (2 days)	Transient	Improvement in nausea and pain; unsuccessful endoscopic dilation; underwent bowel resection
UC								
Pancolitis	20	M	69	PUCAI 60	IMA (100 mg), ileocolic artery (100 mg), right colic artery (100 mg), left colic artery (60 mg), sigmoidal artery (40 mg)	PUCAI 15 (6 days)	>1 month	Improvement in abdominal pain and hematochezia; transitioned to ustekinumab from vedolizumab and infliximab
Pancolitis	22	F	55	PUCAI 50	sigmoidal artery (125 mg)	PUCAI 50 (2 weeks)	1 week	Improvement in diarrhea; transitioned to tofacitinib from vedolizumab and adalimumab
Pancolitis	17	M	64	PUCAI 65	sigmoidal artery, superior rectal artery (450 mg total)	PUCAI 30 (3 days)	Remains in remission	Improvement in diarrhea, improvement in hematochezia; transitioned to tofacitinib from infliximab

CD: Crohn’s disease. UC: ulcerative colitis. SMA: superior mesenteric artery. IMA: inferior mesenteric artery. PCDAI: pediatric Crohn’s disease activity index. PUCAI: pediatric ulcerative colitis activity index. * *p* = 0.025, paired t-test of pre- and post-procedure PCDAI.

**Table 2 jcm-12-02386-t002:** Clinical characteristics of patients with Crohn’s undergoing imaging.

Patient	Age at CD Diagnosis (y)	Sex	Age at Imaging (y)	PCDAI Score at Imaging	Montreal Classification, Disease Location	Montreal Classification, Disease Behavior	Predominant Areas of Inflammation on MRI
1	12	F	17	37.5	L4	B2	Proximal and distal terminal ileum
2	3	M	10	37.5	L2	B2	Sigmoid and upper rectum
3	17	F	17	35	L3	B2	Terminal ileum
4	16	M	17	27.5	L2	B1	Upper and middle rectum
5	14	M	19	20	L2	B2	Proximal terminal ileum

CD: Crohn’s disease. PCDAI: Pediatric Crohn’s disease activity index.

## Data Availability

Pubically available transcriptomic datasets were obtained from the Gene Expression Omnibus (https://www.ncbi.nlm.nih.gov/geo/). The imaging data presented in this study are available on request from the corresponding author. The data are not publicly available to protect patient privacy.

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
