# Peer review of "Case Series of Precision Delivery of Methylprednisolone in Pediatric Inflammatory Bowel Disease: Feasibility, Clinical Outcomes, and Identification of a Vasculitic Transcriptional Program"

_jcm, 2023, doi:10.3390/jcm12062386_

Round 1
Reviewer 1 Report
Journal JCM (ISSN 2077-0383)
Manuscript ID: jcm-2207673
Type: Communication
The manuscript entitled with "Precision delivery of methylprednisolone into inflamed regions of bowel as a bridge therapy in pediatric inflammatory bowel disease" by Steven Levitte et al. as a communication has been recently submitted to the journal JCM. The authors reported that locoregional intra-arterial steroid delivery was conducted directly into specific segments of the gastrointestinal tract to avoid systemic steroid exposure, thereby in this study maximizing tissue concentration while restricting adversely side effects on nine pediatric patients.
As the authors gave the title of “precision delivery of methylprednisolone into inflamed regions of bowel as a bridge therapy in pediatric inflammatory bowel disease”, the related results were limited and shown only one figure. However, the title seems disable to cover the contents in the manuscript. As you may notice, the authors also identified that patterns of vascular morphologic changes are indicative of a vasculopathy within the mesenteric circulation of inflamed segments of bowel in pediatric patients with CD. The authors next performed the single-cell expression analysis and correlation of 60 differentially-expressed vasculitis-associated genes in endothelial cells of patients with active CD who required bowel resection. Both mentioned above were not presented in the title of the manuscript. The reason and the connection for performing two types of analyses after but not before the description of the delivery of methylprednisolone, was not introduced firstly.
Overall, the manuscript was well written but not clearly organized. The results were not nicely presented except the Figure 2. The manuscript was not well focused on the title. In this manuscript, the authors wanted to include more correlations and single-cell expression analysis than the precision delivery of methylprednisolone to treatments of pediatric inflammatory bowel disease. Herein, some concerned issues should be carefully addressed.
Minor Issues:
1) I have no doubts about the interesting results of loco-regional intra-arterial steroid delivery as authors shown in Figure 1. But the efficacy and effective ratio of the therapy via “locoregional delivery of methylprednisolone” were based on only 9 pediatric patients. This cannot be solid though the authors mentioned as the study limitations in the discussion. Such effective therapy must be of great applications but in need of investigations with a large sample size.
2) Supplemental Figure 1. The contents were not too much. I suggest the authors to revise the schema or describe in phrases or suitable form instead. The current form was poor.
3) Supplemental Figure 2. Correlation between PCDAI and vasculitis scores. This figure should be moved back to the main text in order to enhance the major results. I may suggest the authors to replot this figure and set up the proper size to be able to fit into the manuscript.
4) The statistical analysis of the Table 1 was missing.
5) In the “Results” section, there were only two subtitles “clinical outcomes” and “vascular findings”, really? However, the following contents were not logic and could not simply belong to clinical vascular findings. I guess here the authors need two more subtitles to lead the two predictive analyses and correlations.
6) Figure 2A-2C was not properly mentioned in the manuscript. More importantly, the legends to Figure 2 was missing. Hard to understand the Figure 2. It is also not friendly to readers.
7) The authors did create interaction maps (Figure 2B & 2C) to understand the molecular pathways implicated in the differentially expressed genes in active disease v.s. inactive disease (Figure 2A). But the logic was vague. What is the relation between the “Transcriptomic/single-cell analysis” and “the loco-regional intra-arterial steroid therapy”? Will the locoregional delivery be combined with pharmaceutical treatment based on the better understanding of the interaction maps?
8) If the author wanted to highlight the implication of a subset of vasculitis-associated genes as being differentially expressed in the intestinal endothelium in active CD, the title of the manuscript should be nicely modified.
9) Both “loco-regional” and “locoregional” were used throughout the manuscript.
10) Well, “intraarterial” and “intra-arterial” were also mixed and used in the manuscript.
11) Line 119, “mins” could be revised to “min”
12) The data for supporting loco-regional intra-arterial steroid delivery were truly not many but inadequate. No further biochemical analysis and molecular validation were co-support the success of such therapy with regained functions but without any side effects.
13) From Line 218 to Line 249, there was no subtitle for the following paragraphs but clearly the authors talked about different stuffs.
14) In “Discussion” section, Line 259, the “nay” should be corrected to “any”.
15) Figure 2 was telling the “Transcriptomic and single-cell analysis of vasculitis and angiogenesis pathways in Crohn’s disease”, and then, what novel thoughts can be drawn from such predictive analysis and correlations to direct the clinical locoregional therapy?
Since the high-efficacy outcomes of the therapy is truly meaningful, I may recommend this manuscript a Minor Revision can be made.
Reviewer 2 Report
This study evaluated the efficacy of Intra-arterial steroid treatment in 9 pediatric patients with inflammatory bowel disease (IBD). The authors are currently undergoing a clinical trial of Intra-arterial high-dose steroid therapy to treat flares in patients with IBD (ClinicalTrials.gov Identifier: NCT05587673). Therefore, I think this study might be considered as a pilot study. I wonder what the authors think about it. If authors agree that this study is pilot study, authors mentioned about it in the title, abstract, and discussion part.
Most interesting point of this study was described the characteristics of vascular finding of MRE and angiography, despite the small sample size.
The main outcome was to assess the feasibility and efficacy of intraarterial steroid administration. Since the number of enrolled patients was small, a detail description of clinical outcomes should be important.
The authors should describe the followings:
1) Pre-treatment PCDAI
2) Time to assess post-treatment clinical improvement
3) PCDAI score at post-treatment assessment point
4) Duration of durable response of Intra-arterial steroid treatment
5) Combined stricture at the region the steroid injection and the response of stricture
The high-dose steroid was injected intra-arterially; however, since it was administered only once, there will be no adverse effects related to steroid. Instead, angiography is an invasive procedure. I wonder if there are any complications associated with angiography and intra-arterial steroid injection.
Transcriptome analysis is interesting, but it is far from the aim and results of this study. I think this analysis may be moved to the discussion part.
Round 2
Reviewer 2 Report
The authors answered my question appropriately.